# Performance Evaluation of Enhanced ConvNeXtTiny-based Fire Detection System in Real-world Scenarios

**Taimoor Khan, Hacı İsmail Aslan & Chang Choi**
Department of Computer Engineering
Gachon University
Seongnam-si, Gyeonggi-do 13120, KR
`taimooricp@gmail.com,{aslan,changchoi}@gachon.ac.kr`

## Abstract

Timely detection of fires is crucial for saving human lives and minimizing the economic and ecological impact of such incidents. Although numerous attempts have been made to identify a fire in its early stage, significant challenges remain in achieving accurate and reliable detection. Therefore, we proposed a modified pre-trained ConvNeXtTiny architecture for detecting fire, offering high detection accuracy and fast inference time compared to other alternatives over benchmarks. Our source code of the paper will be publicly available at `https://github.com/TaimoorKhan561/ICLR_Source`.

## 1 Introduction

In recent decades, domain experts have increasingly focused on fire detection due to their significant impact on human lives, the economy, and the environment, whether caused by natural or man-made disasters (Khan et al., 2022). To improve the effectiveness of early fire detection, numerous techniques have been developed and evaluated using various datasets and evaluation metrics. For instance, Yar et al. (2021) proposed a novel lightweight Convolutional Neural Network (CNN) inspired by VGG16 for early fire detection and utilized Foggia's and their newly created datasets for experiments. A subsequent study (Yar et al., 2022), developed the Dual Fire Attention Network (DFAN) for real-time indoor and outdoor fire detection, and compare their results with other State-of-the-art (SOTA) techniques using different datasets. Furthermore, Khan et al. (2022) introduced SE-EFFNET for fire scene detection, which achieves a minimized error rate compared to recent literature. Despite the promising results achieved by existing fire detection methods, their detection accuracy and high computation complexity raise questions about their practical implementation in real-time scenarios. To overcome the limitations of existing fire detection methods, we have fine-tuned the ConvNeXtTiny architecture to enhance the accuracy of fire detection while also reducing execution time. Our approach leverages the lightweight nature of the ConvNeXtTiny architecture to achieve real-time performance and improve the overall effectiveness of fire detection. By incorporating fine-tuning strategies, we have improved the accuracy and efficiency of our proposed network, making it a promising solution for real-world fire detection scenarios.

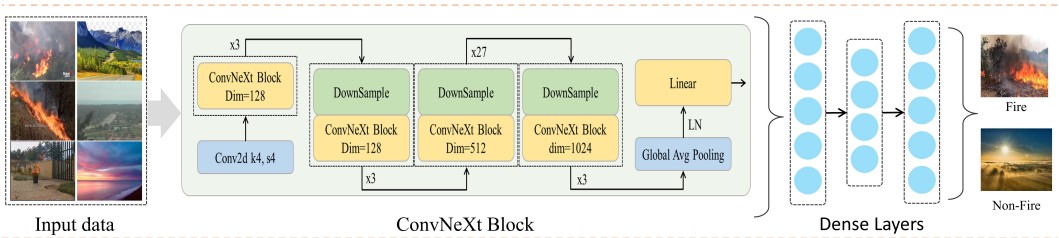

Figure 1: Proposed framework for fire detection with model detail definition.

## 2 METHODOLOGY

In the proposed method, the modified ConvNeXt architecture is leveraged as a backbone model for fire detection. The ConvNeXt model is famous for image classification tasks, achieving top-1 accuracy of 87.8% over the ImageNet dataset (Chen et al., 2023). In this study, a ConvNextTiny is fine-tuned by incorporating the techniques used in the Swin Transformer, which involve downsampling, activation function, inverted bottleneck, and depth-wise convolution that enhance the performance of the proposed ConvNextTiny for fire detection (Liu et al., 2021).

In our approach, we have employed four types of blocks in the ConvNeXt model, which are designed based on the channels in the output feature map, as shown in Figure 1. Each block takes different amounts of time to learn features, and the final prediction of the block becomes the halting point of the original. The output dimension is then doubled under the action of convolution (Li et al., 2022). By incorporating these fine-tuning strategies, we have optimized the ConvNeXt architecture for fire detection, achieving high performance and minimal false prediction rates.

## 3 RESULTS AND EXPERIMENTS

This section presents a comparison between a modified proposed ConvNeXtTiny architecture and other SOTA models over Foggia's and Yar's datasets. Various evaluation metrics such as precision, recall, F1-score, accuracy, False Negative Rate (FNR), and False Positive Rate (FPR) were used to assess the performance of the models. Detailed information about the evaluation metrics and datasets is given in Appendix 1 and 2.

**Comparative Analysis With Baselines** Table 1 presents a comparison of the proposed model with other SOTA methods on two different datasets, namely Yar (Yar et al., 2021) and Foggia's (Foggia et al., 2015). For the Yar dataset, the proposed model outperformed other methods in terms of precision 99.2%, recall 98.7%, F1-score 98.4%, and accuracy 98.5%. It also had the lowest false positive rate of 0% and a false negative rate of 0.12% among all the methods. The DFAN had the second-highest performance with a precision, recall, F1-score, and accuracy of 94.5%, while the SE-EFFNet results were not available for this dataset. For the Foggia's dataset, the proposed model had the highest accuracy 99.91% among all the methods. The DFAN had the highest precision of 98%, recall at 97%, and F1-score of 97%, but the proposed model further improve the performance. The SE-EFFNet had the lowest false positive rate of 0.042% and false negative rate of 0.034% among all the methods. Overall, the results suggest that the proposed model performed better than other baseline architectures on Yar's dataset, while it had the highest accuracy on the Foggia's dataset. These findings demonstrate the effectiveness of the proposed model for fire detection.

Table 1: Comparison of the proposed model with other SOTA methods

| | Yar dataset | | | | Foggia's dataset | | |
|---|---|---|---|---|---|---|---|
| Model | Precision | Recall | F1-score | Accuracy | FPR | FNR | Accuracy |
| ResNetFire (Sharma et al., 2017) | 0.88 | 0.86 | 0.86 | 0.8667 | - | - | - |
| CNN (Yar et al., 2021) | 0.945 | 0.945 | 0.945 | 0.945 | 0 | 0.92 | 0.9715 |
| SE-EFFNet (Khan et al., 2022) | - | - | - | - | 0.042 | 0.034 | 0.972 |
| DFAN (Yar et al., 2022) | 0.98 | 0.97 | 0.97 | 0.97 | 0 | 0.58 | 0.996 |
| **Proposed ConvNeXtTiny** | **0.992** | **0.987** | **0.984** | **0.985** | **0** | **0.12** | **0.9991** |

## 4 CONCLUSION

In this study, we proposed a novel ConvNeXtTiny-based model for effective fire detection in a real-world scenario. The proposed model is evaluated on two benchmark Foggia's and Yar's datasets and achieved state-of-the-art performance. The experimental results showed that the proposed model outperformed baseline methods with an accuracy of 99.91% on Foggia's and 98.5% on Yar's datasets. The proposed model can potentially be used in real-world scenarios for early fire detection and prevention.

ACKNOWLEDGMENTS

This work was supported by the National Research Foundation of Korea (NRF) grant funded by the Korea government (MSIT) (2021R1A2B5B02087169) and under the framework of an international cooperation program managed by the National Research Foundation of Korea (2022K2A9A1A01098051).

URM STATEMENT

The authors acknowledge that at least one key author of this work meets the URM criteria of ICLR 2023 Tiny Papers Track.

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

## A  PERFORMANCE MEASUREMENT METRICS

The following criteria are utilized to evaluate the performance of the proposed model, where TP is true positive, TN is true negative, FP is false positive, and FN is false negative.

$$precision = \left(\frac{TP}{TP + FP}\right) \qquad recall = \left(\frac{TP}{TP + FN}\right)$$

$$\textit{F1-score} = 2 * \left(\frac{precision * recall}{precision + recall}\right) \qquad accuracy = \left(\frac{TP + TN}{TP + TN + FP + FN}\right)$$

$$FPR = \left(\frac{FP}{TN + FN}\right) \qquad FNR = \left(\frac{FN}{TP + FP}\right)$$

## B  ABOUT DATASETS

Extensive experiments are conducted using two publicly available datasets for instance Foggia's and Yar's datasets. Foggia's dataset is a publicly available benchmark dataset that includes 31 videos, gathered from both indoor and outdoor environments containing normal and fire scenes. For experimental results, we extracted 62,690 images from both indoor and outdoor videos for the learning process. In addition, we also utilized Yar dataset, which consists of two classes such as fire and non-fire. Overall, the dataset contains 2000 images for both classes, where both classes have the same size in terms of the number of images.

## C  ENVIRONMENTAL SETTINGS

The proposed ConvNeXtTiny is implemented in Python 3.7 using an Intel Core i9 CPU (Central Processing Unit) (3.60 GHz) and an NVIDIA GeForce RTX 2090 ti GPU (Graphics Processing Unit) with 32 onboard memory in Keras deep learning framework with TensorFlow backend.

## D  COMPLEXITY MEASURES

Alongside high performance and low error rate in the computer vision domain, achieving lower time complexity is also a challenging task for real-world decision making. Therefore, we evaluated and compared the proposed ConvNeXtTiny with two different state-of-the-art architectures such as ResNetFire (Sharma et al., 2017) and DFAN (Yar et al., 2022) in terms of accuracy, inference time, and the number of parameters. In this paper, the experiments were conducted using two different hardware platforms including CPU and GPU as given in Table 2. Comparatively, the ResNetFire achieved smaller Frame Per Second (FPS) rates for both CPU and GPU, which were 2.4 and 57.3 FPS rates, respectively, with a model size of 98.00 MB. In addition, the DFAN obtained high FPS rates of 12.90 for CPU and 70.55 for GPU with a model size of 83.63 MB as given in Table 2, although, these models have a greater size than the proposed model. On the other side, the proposed ConvNeXtiny has a smaller size, which achieved comparatively optimal performance, and also increased the FPS for both CPU and GPU, which are 13.40 and 71.20 FPS, respectively, as mentioned in Table 2. As observed in the experiments, the ResNetFire and DFAN models are not capable of real-world implementation due to their low performance, and smaller inference time in terms of FPS. This achievement enables the proposed ConvNeXtTiny model to be easily deployable over resource constraints devices for real-time decision-making.

Table 2: Time Complexity between proposed ConvNeXtTiny with other SOTA methods on Yar's dataset.

| Model | Frame per second (FPS) | | Parameters (Million) | Model size (MB) | Accuracy (%) |
|---|---|---|---|---|---|
| | CPU | GPU | | | |
| ResNetFire | 2.4 | 57.3 | - | 98.0 | 86.67 |
| DFAN | 12.90 | 70.55 | - | 83.63 | 97.50 |
| ConvNeXtTiny | 15.70 | 78.05 | 6.4079 | 73.76 | 98.50 |

