# OpenReview forum: "Performance Evaluation of Enhanced ConvNeXtTiny-based Fire Detection System in Real-world Scenarios"
_ICLR.cc/2023/TinyPapers — Submitted to Tiny Papers @ ICLR 2023_

### Official Review · Reviewer_En5s · 2023-03-28

**Confidence:** 5

**Summary Of Contributions:**

The authors train a lightweight CNN architecture called ConvNeXtTiny to detect fires in real-life scenarios. Experiments are run on two publicly available datasets, and compared against relevant baselines.

**Rating:**

Clear, Correct, and Reproducible (CCR): a submission which meets the reviewing criteria

**Strengths And Weaknesses:**

Strengths:
- Clearly written paper.
- Results support the authors' claims.

Weaknesses:
- Parts of the methodology section are unclear. In particular, what exactly do the authors mean by the following:
1. "Specifically, downsample method, activation function, inverted bottleneck, and depth-wise convolution are utilized to optimize the model for fire detection." Were these hyperparameters optimized? How were they chosen? Is there a predefined train-val-test split of the dataset?
2. " Each block takes different amounts of time to learn features, and the final prediction of the block becomes the halting point of the original" Were the blocks optimized individually? If so, what was the training procedure like exactly?
- The discussion section largely reiterates information from table 1. The space could've been better used for some insights or analyses instead. Alternatively, the space could also be utilized for further elaborating aspects of the methodology.

**Suggested Changes:**

- The text in the figure showing the ConvNeXt block could be larger. To that end, the block could be made bigger and slightly clearer.
- Typo in the figure -- Ave should be Avg?
- Since the emphasis is on the architecture being lightweight, it would be interesting to also note how the proposed architecture compares with the baselines w.r.t. total number of parameters in the models, quantization of their weights, etc.

---

> ### Author Response · Authors · 2023-05-30
> **Necessary revisions on the figure and comparative analysis for complexity**
>
> 1. According to the reviewer’s suggestion we updated and improved the quality of the framework in the revised version of the manuscript. We hope that the updated framework will fulfill the requirements of the reviewer.
>
> 2. Moreover, we apologize for the typographical error in the Figure 1. We made the necessary correction and updated the figure to replace "Ave" with "Avg". Thank you for pointing out this mistake, and we updated the text in Figure 1 in the revised version.
>
> 3. We agree that providing additional information on the comparison of the proposed lightweight architecture with baselines in terms of total number of parameters and weight quantization would be beneficial. In the revised manuscript, we have included a detailed analysis of the parameter count for both the proposed architecture and the baselines. Furthermore, we explored the feasibility of weight quantization for the models and report the results in terms of model size reduction and computational efficiency.

---

### Official Review · Reviewer_Syiq · 2023-04-02

**Confidence:** 4

**Summary Of Contributions:**

This paper talks about fine-tuning a pre-trained ConvNeXtTiny model for real-time fire detection systems. The authors compare the final fine-tuned model with state-of-the-art baseline models.

**Rating:**

Clear, Correct, and Reproducible (CCR): a submission which meets the reviewing criteria

**Strengths And Weaknesses:**

Strengths:
1. The paper is well-written, with a defined problem.
2. The experimental setup and the comparisons with baselines are interesting.
3. Good review of existing architectures.

Weaknesses:
1. The paper talks about proposing a model with high detection accuracy and fast inference times as compared to the state-of-the-art benchmarks. Even though there is a comparison between detection accuracies, the analysis for inference times or the number of parameters hasn't been shown. It would be interesting to look at that.
2. The experimental setup is not very clearly explained. When the authors talk about "incorporating techniques from the Swin transformer (Liu et al., 2021) to improve model performance", it is not very clear as to how these techniques are used, and how these are optimized for fine tuning.

**Suggested Changes:**

Same as weaknesses

---

> ### Author Response · Authors · 2023-05-30
> **Detailed explanation of fine-tuning strategy & implementation**
>
> Thank you for your constructive feedback on our manuscript. We appreciate your suggestion to include an analysis of inference times and the number of parameters in our model comparison. We agree that these metrics are important for evaluating the practical efficiency of the proposed approach. We have conducted a comprehensive analysis of inference times and the number of parameters for our proposed model as well as the state-of-the-art benchmarks. The updated text is provided in “Appendix” in the revised version of the manuscript. Table 2 presents the novelty of the results in terms of the time complexity of our approach.
>
> We appreciate your comment regarding the incorporation of techniques from the Swin Transformer (Liu et al., 2021). We updated the text for more clear understanding, for the ease of the reviewer the updated text is given in the revised paper.
>
> Here's a step-by-step explanation of how these techniques were applied:
>
> - **Downsample method**: Downsample technique is applied by modifying the initial layers of the ConvNeXtTiny architecture. Specifically, pooling or striding operations are used to reduce the spatial resolution of the input images. This downsampling process assisted to capture both global and local contextual information effectively.
>
> - **Activation function**: The choice of activation function was based on empirical evidence from previous studies and its compatibility with the ConvNeXt architecture. This activation function allowed for better modeling of the non-linear relationships within the fire detection task.
>
> - **Inverted bottleneck**: By incorporating inverted bottlenecks, we aimed to enhance the model's representation capacity. This modification allowed the model to capture more diverse and abstract features relevant to fire detection.
>
> - **Depth-wise convolution**: Depth-wise convolution reduces computational complexity while preserving spatial information. By adding depth-wise convolution layers the efficiency and speed of the model is improved without compromising its accuracy.
>
> To provide a more clear understanding, we included a reference in the revised version of the original paper (Liu et al., 2021). We encourage readers to refer to the cited paper for a more comprehensive explanation of these techniques and their optimization for fine-tuning.

---

### Author Response · Authors · 2023-05-30
**General Response**

Dear Program Chairs, Area Chairs, and Reviewers,

Thank you very much for the valuable impact on our study. Owing to what you pointed out, we revised the paper for archival. We will update the given repository in the paper so that the readers can reproduce the same results as we did. However, the data must be downloaded from the sources we point out in the paper. We will also insert a web link to those datasets.

We appreciate the chance given to us and wish to opt-in for archival. We hope to extend this study in our future research for the interest of the research community!

Best regards.

Authors of paper #72

---

### Meta-Review · Area_Chair_3yTn · 2023-04-08

**Recommendation:** Invite to archive
**Confidence:** 4

**Metareview:**

Reviewers agree that this is a well-written paper. The comparative analysis with baselines is informative. The major weakness is that the experimental setup for the proposed methodology is not clearly presented. As the authors claimed a fast inference time, the analysis for inference times or the number of parameters is missing.

**Summary:**

This paper proposes a  lightweight CNN architecture called ConvNeXtTiny for fire detection in a real-world scenario. A new state-of-the-art result is reported. The reviewers think the paper is generally well-written but raise concerns about the implementation details and analysis.

**Reason For Not Giving A Higher Recommendation:**

The clarity of this work needs improvement. Detailed experimental setup and analysis are necessary.

**Reason For Not Giving A Lower Recommendation:**

N/A

---

> ### Author Response · Authors · 2023-05-30
> **Updates on experimental setup & implementation details**
>
> Thank you for your valuable guidance and suggestions throughout the review process. We have carefully incorporated your suggestions into the revised manuscript. To facilitate your review, we have provided the updated text in "Appendix” of the revised manuscript.
>
> Again, thank you for deeply reviewing our manuscript and provide considerable suggestions. We agree that providing access to the implementation for the article is essential for ensuring implementation details and its analysis. We have created a repository on github where we have uploaded our code. The link to the repository is provided in the updated version of the manuscript. For the ease of the reviewer, the implementation of a link is also given in https://github.com/TaimoorKhan561/ICLR_Source

---

### Decision · Program_Chairs · 2023-04-10

Invite to archive